# Petrophysics-guided velocity analysis and seismic data reprocessing to improve mineral exploration targeting in the Irish Zn-Pb Orefield

Victoria Susin<sup>1,2</sup>, Aline Melo<sup>1,2</sup>, Koen Torremans<sup>1,2</sup>, Juan Alcalde<sup>3</sup>, David Martí<sup>3</sup>, and Rafael Bartolome<sup>4</sup>

<sup>1,2</sup> School of Earth Sciences and Research Ireland Centre for Applied Geosciences (iCRAG), University College Dublin, Belfield, Dublin, Ireland

10 Correspondence to: Victoria Susin (victoria.figueirasusin@ucdconnect.ie)

Abstract. The Limerick Syncline, part of the Irish Zn-Pb Orefield in southwest Ireland, represents a geologically complex and relatively underexplored region, despite hosting the Stonepark and Pallas Green Zn-Pb deposits. The mineral deposits in the Syncline are largely stratabound Zn-Pb systems hosted within Mississippian carbonates. In the area, a thick volcanic sequence overlies and interfingers with the carbonate host rocks, mineralisation and alteration. This has posed significant challenges to seismic imaging in the region, resulting in a poor understanding of the overall structural setting. This study presents an optimised seismic processing workflow tailored to these geological complexities and applied to a 2D seismic reflection profile. The workflow integrates information from newly acquired downhole and laboratory P-wave velocity data with first-arrival travel-time tomography to produce a new velocity model for post-stack migration. This resulted in better signal recovery and enhanced reflector coherence, in particular, reflection continuity. As a result, imaging of key stratigraphic boundaries, internal form lines and the lateral interfingering of volcanic and carbonate units was enhanced. Acoustic impedance analysis using laboratory density data enabled a better understanding of the origins of seismic reflectivity and a more confident geological interpretation of the laterally variable lithologies. A chaotic, low-amplitude seismic facies was recognised representing laterally persistent breccia corridors which may provide a practical indirect seismic proxy for significantly hydrothermally altered zones in the carbonates. Critically, two major previously unrecognised basin-scale faults were identified to the south of the Stonepark and Pallas Green deposits, bounding a significant (half-)graben. Thickness patterns and igneous packages indicate late Tournaisian to early Viséan syn-depositional faulting coeval with emplacement of the Limerick Igneous Suite, with subsequent Variscan inversion providing a net-zero displacement at the surface. These results expand the exploration research space beyond the known mineralisation areas, especially around normal faults on the southern flank of the Syncline.

#### 30 1 Introduction







The transition to a low-carbon society has significantly increased the demand for essential resources in recent years, driving higher consumption of base metals and critical materials needed for advancing sustainable technologies (Vidal et al., 2013; Ali et al., 2017). Meanwhile, the depth to target across the mining industry continues to increase globally (Alcalde et al., 2022). Specifically in Europe, the recent Critical Raw Materials Act (CRMA) has reinforced the need for sustainable sourcing of critical raw materials within the European Union (Hool et al., 2024), requiring increasingly comprehensive analysis of the subsurface resources. This incentivises the development of cost-effective, more sustainable and high-resolution geophysical methods for mineral exploration (Malehmir et al., 2020; Alcalde et al., 2022).

Due to the relatively limited resolving power of traditional mining geophysical methods, such as electromagnetics, induced polarisation, and potential field techniques, at greater depths, the adoption of methods from the oil and gas industry has gradually become more common in mineral exploration (Eaton et al., 2003). The seismic reflection method can provide high-resolution images of the subsurface with a greater penetration depth, complementing and defining target areas for drilling

<sup>&</sup>lt;sup>3</sup> Geosciences Barcelona GEO3BCN, Barcelona, Spain

<sup>&</sup>lt;sup>4</sup> Instituto de Ciencias del Mar (ICM) CSIC, Barcelona, Spain

campaigns, which has led to the growing interest in seismic methods as a mineral exploration tool (Malehmir et al., 2012). Its relatively recent success in imaging and characterising mineral prospects and their structural controls, depth extensions, and host rock properties has been highlighted in multiple studies (e.g., Eaton et al., 2010; Manzi et al., 2019; Malehmir et al., 2020; Gil et al., 2021; Cheraghi et al., 2023; Bell et al., 2023).






In Ireland, 2D seismic surveys have been conducted to better understand the structural and stratigraphic setting of the Irish Mississippian basins to assist exploration for Irish-type Zn-Pb deposits (e.g., de Morton et al., 2015; Ashton et al., 2018). These deposits are hosted within hydrothermally altered carbonate rocks and are predominantly associated with normal faults in relay-ramp systems (Wilkinson et al., 2015). Spanning an area of approximately 40,000 km², the Irish Orefield has been a significant focus of exploration. Since the 1950s, five deposits, including Lisheen, Galmoy and the world-class Navan deposit, have been developed into producing mines (Blaney and Redmond, 2015).

Ashton et al. (2018) reported the successful application of seismic reflection surveys to the discovery of the Tara Deep deposit at depths of 1,500 meters in the vicinity of the Navan deposit. Currently, approximately 1,000 km of 2D reflection seismic exists across the Irish Midlands.

As part of mineral exploration efforts, six 2D seismic reflection profiles were acquired in 2011 near the Stonepark and Pallas Green Zn-Pb deposits, on the northwestern edge of the Limerick Syncline, in southwest Ireland (Fig. 1). The Stonepark deposit has a resource of 5.1 Mt grading 8.7% Zn and 2.6% Pb at depths of 200 meters, and the Pallas Green deposit, a resource of 45.4 Mt grading 7% Zn and 1% Pb at depths between 200 and 1,000 meters (Gordon et al., 2018; Blaney and Coffey, 2023).

In this area, the carbonate rocks that host the sulphide mineralisation are intruded by and interbedded with a thick volcanic succession (Blaney and Coffey, 2023), whose distribution is still poorly understood. Volcanic rocks pose difficulties for seismic imaging because of signal attenuation (Liberty et al., 2015; McBride et al., 2021) and scattering, caused by abrupt vertical changes in elastic properties (e.g., Pujol et al., 1989; Planke et al., 1999; Ziolkowski et al., 2003; Sullivan et al., 2011). These effects have discouraged further seismic exploration in the target area due to the low resolution obtained in previous imaging efforts.

The effectiveness of seismic methods is highly site- and geology-dependent (Malehmir et al., 2012). In geologically complex settings such as the Limerick Syncline, characterised by abrupt lateral facies changes (Somerville et al., 2011), effective seismic processing with suitable velocity models and migration algorithms is crucial to achieve a reliable seismic image (Singh and Malinowski, 2023). Additionally, lithological and petrophysical characteristics of the stratigraphic units, especially the volcanic rocks, must be considered during data processing to ensure accurate imaging and preserve true amplitude responses (Planke et al., 1999). However, suitable velocity characterisation and well-constrained petrophysical data are currently missing for the Irish Midlands, especially for the Limerick Syncline. Furthermore, a robust interpretation of the acoustic impedance variations and of the origin of seismic reflectivity remains absent in this part of the basin.

Therefore, to address these data gaps and improve seismic imaging, we developed and implemented a reprocessing workflow on the profile LK-11-02 to enhance data interpretability in the Limerick Syncline (Fig. 1). The objective of this work is to improve subsurface characterisation by revisiting mineral exploration seismic datasets and integrating newly acquired information. The new processing sequence incorporates information from recently acquired downhole and laboratory petrophysical data, along with first-arrival travel-time tomography, which provided essential insights into the velocity field.

We present a velocity analysis exploring the petrophysical characteristics of the main units in the Limerick Syncline and how the laboratory P-wave velocity correlates with the travel-time tomography. This analysis was used to inform the construction of the stacking velocity model, leading to an improved seismic image with enhanced resolution. The workflow enabled interpretation of the main lithologic boundaries and fault zones, highlighting the potential of seismic imaging in subsurface characterisation and its implications for mineral exploration in the area.

Figure 1: Geological map of the Limerick Syncline 1:100,000 (Geological Survey Ireland, 2022) showing the seismic lines with profile LK-11-02 highlighted in red, the drill holes used for downhole and laboratory petrophysical measurements (green circles) and the location of the Stonepark and Pallas Green prospects. The inset shows the location of part of the Limerick Syncline study area (black rectangle) within Ireland.

#### 2 Geological setting



The Limerick Syncline hosts a stratigraphic sequence consisting of siliciclastic, carbonate and volcanic rocks (Somerville et al., 1992; Fig. 2). The Lower Palaeozoic basement, underlying the basinal succession, consists of greywackes, shales and volcanic rocks deformed and weakly metamorphosed during the Caledonian orogeny (Chew et al., 2009). Lying unconformably atop the basement are largely Devonian continental red bed conglomerates, sandstones, and mudrocks of the Old Red Sandstone Formation, likely deposited in grabens. A northward-directed marine transgression across Ireland occurred during the early Mississippian, depositing progressively deeper marine carbonates (Sevastopulo and Wyse Jackson, 2009). The oldest transgressive unit is the Lower Limestone Shale Group (LLS), comprising a heterolithic series of interbedded shales, siltstones and sandstones, and carbonates (Somerville and Jones, 1985; Tyler, 1997; Blaney and Redmond, 2015). A transition to fully marine sedimentation followed in an open carbonate ramp environment, consisting of a sequence of argillaceous bioclastic limestones (ABL) of the Ballymartin Formation and Ballysteen Formation, together forming the Ballysteen Group.

Finally, increasing water depths allowed the development of laterally extensive sequences of micritic mud mounds of the Waulsortian Limestone Formation (WAL; Somerville and Jones, 1985).



Above the WAL lies the Lough Gur Formation (LGG), a cherty, bioclastic limestone, succeeded by the Knockroe Volcanic Formation (KKR), part of the Limerick Igneous Suite and comprising a series of basaltic lavas and pyroclastic strata variably altered and brecciated (Strogen, 1988; Somerville et al., 1992; Blaney and Coffey, 2023; Slezak et al., 2023). The KKR is interfingered with the carbonate beds of the LGG and the overlying Herbertstown Limestone Formation (HBN), dominated by grainstones and packstones of shallow-water origin (Somerville et al., 1992). Towards the top, HBN is interbedded with the Knocksheefin Volcanic Formation (KKS), the second volcanic unit of the Limerick Igneous Suite, consisting of variably altered basanites and tuffs. Chlorite-clay alteration is typically seen in both KKR and KKS (Slezak et al., 2023). The interfingering of volcanics and carbonates forms a package up to 900 meters thick in the Limerick Syncline. These units are cut by later porphyritic basalt dykes and sills (Strogen, 1983; Slezak et al., 2023).

Figure 2: Stratigraphic column of Limerick Syncline (adapted from Somerville and Jones, 1985; Strogen, 1988; Somerville et al., 1992; Blaney and Redmond, 2015). Brecciated zones are drawn in grey, Zn-Pb mineralisation in red and intrusions in pink.

The Waulsortian-hosted Zn-Pb deposits occur largely in the hanging walls of complexly segmented relay-ramp normal fault systems (Hitzman, 1999; Torremans et al., 2018; Kyne et al., 2019). These fault zones are associated with Early Mississippian rifting in the Midlands, having a similar age to the upper ABL and WAL deposition (Johnston et al., 1996; de Morton et al., 2015). Previously, Somerville and Strogen (1992) argued the absence of syn-sedimentary faulting in the Shannon Trough, and, to date, no large ore-controlling faults have been identified in the study area (Blaney and Redmond, 2015). Blaney and Coffey (2023) discuss the association between intrusions in the Limerick Syncline and syn-volcanic faults, and the possibility that these faults were the primary conduits for mineralising fluids. In the study area, most of the mineralisation occurs within the lower half of the WAL as lenses hosted by breccia bodies. Brecciation and mineralisation formed through alteration and precipitation-dissolution processes by hydrothermal fluids (Blaney and Redmond, 2015).

# 3 Data and Methods





This study focuses on petrophysics-guided velocity analysis and the seismic reprocessing of profile LK-11-02, part of a series of 2D surveys acquired on the northwestern edge of the Limerick Syncline (Fig. 1). The workflow (Fig. 3) consists of (i) combining newly acquired on-core petrophysical laboratory measurements, downhole sonic data, and travel-time tomography to produce a velocity analysis that constrains the velocity field of the area, (ii) using this velocity information to guide semblance analysis for stacking velocity construction, and (iii) reprocessing profile LK-11-02, originally processed in 2011 and reprocessed in 2018 (here referred to as datasets 1 and 2, respectively), by incorporating the 2018 products together with the updated velocity model. Additionally, acoustic impedance was derived from the petrophysical properties to provide further constraints for a more comprehensive, geology-driven interpretation of the new seismic image.

Figure 3: Flowchart showing the data and processing workflow used in the study. Dataset 1 and dataset 2 comprise the data made available for the reprocessing. Outputs are the workflow and products developed to improve the seismic image.

# 3.1 Drill hole data


Ten drill holes were studied in detail as part of this work, with drilled depths ranging from 350 to 900 meters (Table 1; Fig. 4). Eight of the drill holes, including TC-2531-001 (Chakraborti et al., 2025), TC-2638-004 (Susin et al., 2025b), TC-2638-008 (Susin et al., 2025c), TC-2638-026 (Susin et al., 2025d), TC-2638-036 (Susin et al., 2025e), TC-2638-074 (Susin et al., 2025f), TC-2638-088 (Susin et al., 2025g), and TC-2638-101 (Susin et al., 2025h), are located near the seismic profile trace.

The remaining two, G11-2531-01 (Melo et al., 2025a) and G11-2638-06 (Susin et al., 2025a), are situated 4.9 and 1.8 km away, respectively (Table 1). Most drill holes intersect thick accumulations of WAL and typically terminate a few meters into the underlying ABL. Six of the ten drill holes also intersect higher stratigraphic units, including the LGG, KKR, HBN, and KKS (Fig. 4).



Figure 4: Lithological sections relative to topography and distance between drill holes. Stratigraphic information is derived from geological logs.

The top of drill hole TC-2638-088 (Fig. 4) intersects the KKR for 95 meters, followed by 70 meters of LGG before reaching the WAL and the top of the ABL. South of TC-2638-088, in the central part of the seismic profile, the drill hole TC-2531-001 is the deepest in the region, providing 926.60 meters of stratigraphic information in the central part of the Limerick Syncline. It crosses a thick, interfingered carbonate-volcanic sequence, intersecting the KKS, transitioning to the carbonates of the HBN, and exhibiting predominantly lavas and tuffs from 420 meters depth until the bottom.

Table 1 Drill hole information for petrophysical measurements and distance to the seismic profile LK-11-02.

| Drill Hole ID | Length (m) | <b>Total Samples</b> | Distance to LK-11-02 |
|---------------|------------|----------------------|----------------------|
| TC-2531-001   | 926.6      | 199                  | 144 m                |
| TC-2638-004   | 440        | 50                   | 430 m                |
| TC-2638-008   | 605        | 51                   | 430 m                |
| TC-2638-026   | 349.6      | 50                   | 120 m                |
| TC-2638-036   | 352.5      | 39                   | 78 m                 |
| TC-2638-074   | 389.5      | 50                   | 20 m                 |
| TC-2638-088   | 677.3      | 50                   | 297 m                |
| TC-2638-101   | 461.3      | 50                   | 195 m                |
| G11-2531-01   | 520        | 84                   | 4,890 m              |
| G11-2638-06   | 518.6      | 88                   | 1,830 m              |

Drill holes G11-2531-01 and G11-2638-06 (Fig. 4), located a few kilometres away from the seismic profile, provide important downhole petrophysical information about the stratigraphic sequence of the north and south flanks of the Syncline. Both drill holes cross the stratigraphy from the LGG to the ABL. G11-2531-01 intersects approximately 300 meters of interfingered volcanic and carbonate rocks before reaching the LGG.

# 3.2 Petrophysics data








Laboratory P-wave velocity (Vp) and density data were acquired for over 700 core samples, covering the full length of the available drill holes. Measurements were conducted on NQ-sized samples, each 10 centimetres in length. The sampling interval was planned based on the total drill hole length, aiming for approximately 50 samples per drill hole, except for TC-2531-001, G11-2531-01 and G11-2638-06, where denser sampling was carried out (Table 1). In addition to the laboratory measurements, drill holes G11-2531-01 and G11-2638-06 include downhole sonic log datasets.

The Vp measurements were taken on dry core samples using a Proceq Pundit Lab ultrasonic pulse velocity device (Melo et al., 2025b). Grain size variations in the samples can influence the pulse propagation. This effect can be minimised by selecting a frequency such that the wavelength is at least twice the size of the largest grains (Proceq Pundit Lab manual). In our study, the samples consist primarily of volcanic and carbonate rocks from the KKR, WAL and ABL (Fig. 4). A frequency of 150 kHz was selected, assuming a wavelength twice as large as the coarsest particles in the samples, such as fossils. Quality control was applied to the dataset, including filtering to retain only measurements with an accuracy greater than 75%.

The density measurements were taken on the same core samples, which were soaked in water for 24 hours before measurement, using an Ohaus Explorer balance equipped with an under-hook weighing system (Melo et al., 2025b). Density was calculated using Archimedes' principle, which consists of weighing the sample in both air and water. The water temperature was considered in the calculation.

The sonic log data was acquired using full-waveform sonic probes. These probes employ dual transmitters and receivers to emit and detect high-frequency acoustic waves, measuring the first-arrival transit time of P-waves through the formation. The precise distance between the transmitters and receivers enables computation of the formation acoustic velocity (Robertson Geo Mining & Minerals, 2019).

#### 3.3 Travel-time tomography

To extend the velocity analysis beyond 1D velocity measurements from drill holes, a tomographic velocity model was obtained from profile LK-11-02, using the PStomo\_eq 3D local earthquake (LE) tomography algorithm (Benz et al, 1996; Tryggvason et al., 2002) that also handles the inversion of controlled source data (together and separately). The model is based on first-arrival picks from dataset 1 used for statics corrections, which were complemented with new travel times manually picked at an interval of 250 meters in CMP gathers for far offsets in this study. Forward modelling was performed by computing the time field from a source or receiver to all the model cells, sized 40 m x 10 m x 5 m (for x, y, z). Then, the travel times to source and receiver positions were calculated from the time field and the ray tracing was performed backwards and perpendicular to the isochrons (Tryggvason and Bergman, 2006). The inversion part of the processing is performed with the conjugate gradient solver LSQR (Least Squares QR; Paige and Saunders, 1982).

An appropriate initial velocity model is essential to ensure convergence during travel-time inversion. In this study, a smooth 2D model was developed using a priori constraints from the petrophysical Vp data and drilling control on formation depths in the area. Initial inversions were performed using simple 1D models with smooth velocity gradients and no lateral variations to assess first-arrival pick quality, which provides the primary constraint for the inversion, and to identify potential acquisition geometry errors. Inaccurate or inconsistent picks can introduce significant errors in the velocity model, hindering convergence

and reducing the reliability of the results. Preliminary inversions, however, showed satisfactory convergence and reduction in misfit by an order of magnitude. Based on these results and supported by the petrophysical data, the initial models were updated to include smooth lateral velocity variations and depth-dependent velocity gradients, reflecting general trends without overfitting and maintaining independence from the initial model. The simplicity of the model was supported by the robustness of the inversion algorithm, which allowed effective use of these generalised velocity trends. Updated models enabled faster inversions, a smaller inversion grid size, and improved resolution of the final velocity model.

# 3.4 Seismic imaging

#### 3.4.1 Seismic data





Profile LK-11-02 consists of a 10.88 km line acquired using a split-spread geometry with a Vibroseis seismic source, featuring 20 m source spacing and 10 m receiver spacing. The sweep ranged from 8 to 160 Hz over 16 s with a taper of 300 ms. Data was recorded with a 1 ms sampling interval and a record length of 3000 ms. Near and far offsets are 5 m and 2995 m, respectively.

The data available in the study consists of two datasets (Fig. 3):

- i. Dataset 1: Raw field data, stack, and post-stack time migration delivered to Teck Resources Limited by Velseis Processing Ltd. in 2011, licensed from the Geoscience Regulation Office Department of Climate, Environment, and Communications of the Government of Ireland.
- ii. Dataset 2: CMP gathers (containing geometry, statics correction, and de-noising), stack, and pre-stack time migration, processed by Earth Signal Processing Ltd. (ESP) in 2018 for Group Eleven Resources, used under license.

These two datasets serve as starting points for our workflow. First-arrival picks extracted from the header of dataset 1 were quality-checked, corrected and complemented for tomographic inversion. CMP gathers from dataset 2 were used to generate the new seismic image.

While seismic pre-processing techniques were applied, this study does not focus on their evaluation. Instead, it emphasises the impact of incorporating additional information about the velocity field, refining the velocity model and improving post-stack processing.

The processing workflow for dataset 2 is described in Table 2. It applies a pre-processing sequence to enhance the signal-to-noise (S/N) ratio in the raw data, a surface-consistent amplitude scaling to address variable source and receiver coupling, and residual scaling to enhance offset contribution and to preserve amplitudes in noisy areas. A set of statics corrections was also applied by ESP, minimising travel-time distortions in the CMP gathers and improving the stack response. The Kirchhoff prestack time migration was applied using 100% of the smoothed stacking velocities.

Table 2 Processing workflow of dataset 2.

1. Geometry and QC

2. Refraction analysis

3. Replacement statics:
 Datum: 0 m, replacement velocity 3500 m/s

4. Trace editing

12. Surface consistent statics
13. Final semblance velocity analysis:
 Picks 250 m interval

14. Coherent noise attenuation: 12 Hz cut-off
15. Spectrum analysis (shots)
16. Mute
17. Sort to CMP gathers

6. Amplitude scaling

7. Vibroseis designature

8. Fourier filter:

0-1000 m/s velocity, 0-28 Hz frequency

9. Deconvolution: 5/8 - 130/150 Hz

10. Residual scaling

11. Semblance velocity analysis 1:

Picks 500 m interval

18. Migration
Kirchhoff pre-STM using 100% of smoothed velocity

19. Spectrum analysis (gathers)

20. Final mute

21. Stack

22. Multichannel trace scaling

23. Filter: 25/40 – 65/75 Hz

It is important to note that full recovery of the original seismic products was not possible. Using two datasets reflects a common challenge in mining, imposing practical limitations on legacy data availability, including incomplete records, missing processing parameters, and limited documentation of prior workflows.

# 3.4.2 Reprocessing workflow






Our seismic processing workflow is outlined in Table 3. It focused on producing a velocity analysis for the area and incorporating the new information into the seismic processing applied in our study. The workflow uses CMP gathers from step 17 in Table 2 as input and includes semblance velocity picking, normal moveout correction (NMO), CMP stacking, de-noising, and post-stack time migration. The stacking velocity model is informed by petrophysical and tomographic results following the normal moveout correction. In qualitative terms, when both laboratory Vp and tomography indicate high velocities in the shallow carbonate units, the stacking velocity is adjusted to reflect these values, aligning both the geological setting and the need to flatten reflectors during the velocity analysis.

The velocities were picked at an interval of 250 m based on the interactive normal-moveout correction and guided by the average velocity model obtained from the tomographic inversion and petrophysical measurements. After quality control of the stack section, we applied an additional de-noising sequence including Butterworth filtering, time-frequency domain (TFD) noise attenuation and 2D spatial filtering to suppress remaining coherent and random noise. The final step in the processing sequence was a Kirchhoff post-stack time migration, using 100% of the smoothed stacking velocity. Intended to relocate dipping events to their correct positions, collapse diffractions, and increase resolution.

Given the complex geology of the study area and the noisy data, we avoided adding unnecessary complexity to the velocity model. This decision was made to prevent potential challenges and instability in the Kirchhoff time migration, particularly when dealing with sharp velocity contrasts (Pu et al., 2021). For the same reasons, we opted for post-stack migration, as prestack migration would require a more detailed velocity model than the data or geology allows.

**Table 3** Reprocessing workflow applied to dataset 2 CMP gathers.

#### Reprocessing workflow

- Semblance velocity analysis Picks 250 m interval
- 2. NMO 30 % stretch mute
- 3. Stack
- 4. Butterworth filter: 12/17 50/100 Hz

- 5. FTD noise attenuation 100 ms window, 5-70 Hz frequency
- 6. 2D spatial filtering
- 7. Migration
  Kirchhoff post-STM using 100% of the smoothed velocity

#### 4 Results





## 4.1 Velocity analysis

P-wave velocities were measured on core samples from drill holes G11-2531-01 and G11-2638-06 and compared with downhole sonic logs to assess the correlation between the two methods. To address the different sampling intervals (1 cm in sonic logs vs ~10 m in core samples), the sonic log curve was resampled to a coarser resolution, anchored at depths corresponding to laboratory measurements (Fig. 5). Both datasets displayed similar Vp trends, although the sonic logs yield slightly higher velocities. These offsets are attributed to differences in measurement conditions (e.g. lithostatic vs. atmospheric pressure), sample state (fresh vs. time-degraded core), pore fluid effects (laboratory measurements were done on dried samples), anisotropy, and instrument frequency responses. Nevertheless, the good correlation between the two datasets provided the confidence to extend our velocity analysis campaign to the remaining eight drill holes. The Vp measurements were used in this study as qualitative geological constraints, validating velocity trends, identifying key velocity contrasts, and informing the velocity picking in ambiguous zones with low semblance coherence.

The laboratory Vp dataset, comprising a total of 640 measurements, was classified by stratigraphic units (Table 4). Because velocity in carbonate rocks is sensitive to porosity and pore type (Eberli et al., 2003), samples were also visually grouped by lithological and textural characteristics. The main lithological categories are *clean* (packstones and grainstones), *bioclastic* (limestones with fossil fragments, typically wackestones) and *mud-rich* (samples dominated by fine fractions, either micrite or clay/argillaceous). Additional textures, such as dolomitization, brecciation and mineralisation, were also included in the classification.

The Vp values of the units in the Limerick Syncline range from 3100 to 6500 m/s (Table 4). Overall, the velocity contrasts within the stratigraphy are very mild. Notable velocity contrasts are seen at the contact between the volcanic-dominated (KKR and KKS) and the carbonate-dominated (LGG and WAL) units, as well as between WAL and ABL (Fig. 5 and Table 4). The carbonate units show higher overall velocities compared to the volcanic rocks. Velocity outliers observed in the carbonate sequence are related to increased mud content, dolomitization, brecciation and mineralisation (as individual textures or combined). The velocities per texture are not examined in detail in this study.

To extend and validate the Vp petrophysical measurements over the entire study area, we performed travel-time tomography, resulting in a P-wave velocity model (Fig. 6). The velocity field was resolved from the surface topography down to a maximum depth of 245 m. The resolution in travel-time tomography is strongly constrained by the available offsets and the seismic velocities involved in the inversion, which determines the final ray coverage used to plot the velocity models. For this reason, new picks for far offsets were included in the inversion, although these were limited because of the low S/N ratio in some of the acquired seismic reflection data.

Table 4 Laboratory P-wave velocities of the stratigraphic units, and main carbonate lithologies and textures.

| Units | Velocity (m/s) | Median (m/s) | Lithology/Texture | Velocity (m/s) | Median (m/s) |
|-------|----------------|--------------|-------------------|----------------|--------------|
| KKS   | 3100 - 5900    | 4800         | Clean             | 5800 - 6400    | 6100         |
| HBN   | 5300 - 6400    | 5900         | Bioclastic        | 5600 - 6300    | 6000         |
| KKR   | 3500 - 6300    | 5600         | Mud-rich          | 5000 - 6200    | 5700         |
| LGG   | 5400 - 6400    | 6100         | Dolomitized       | 3700 - 6400    | 6000         |
| WAL   | 3700 - 6500    | 6000         | Brecciated        | 4600 - 6500    | 5700         |
| ABL   | 5000 - 6200    | 5700         | Mineralised       | 5100 - 6500    | 6200         |

The uppermost layer in the model, outlined by a purple line in Fig. 6, shows a low-velocity zone (purple to blue colours, ≤ 2000 m/s) with a thickness ranging from 10 to 50 m, which, according to core logging data, can be correlated with the overburden. Below the immediate overburden, a thick low-velocity zone (≤ 3500 m/s) appears in the centre of the profile,

Figure 5: Comparison of downhole and laboratory P-wave velocity data a) for drill hole G11-2531-01, and b) for drill hole G11-2638-06. The light grey curve represents the downhole data, the blue curve represents the resampled downhole data, and the red dots represent the laboratory data. The background colours represent stratigraphic units intersected by the drill holes.

south of 146000mN, thickening up to 170 meters close to drill hole TC-2638-088 (arrows 1 and 2; Fig. 6), and reducing in thickness around 144000mN. These lower velocities correspond to altered volcanic rocks from the KKS and KKR units in TC-2531-001 (0 to -60 m) and TC-2638-088 (30 to -60 m). Laboratory measurements for these intervals also reveal reduced velocities between 4700 and 5900 m/s and 4800 and 5400 m/s, respectively.





Zones of highly porous limestones with visual evidence of cavitation are seen from 0 to -60 m in drill holes, indicating greater depths than usual in karst-related features in this area. Karstification is common in the KKR and the limestones of the LGG and WAL, as well as along fault zones (McCormack et al., 2017; O'Connell et al., 2018).

Zones of elevated velocities (4000 to 6500 m/s) are present near the surface on the southernmost side of the profile and below the central low-velocity zone (Fig. 6). Drill hole TC-2638-101 shows comparably elevated laboratory velocities (6000 to 6300 m/s) between 70 to -100 m (arrow 3; Fig. 6), corresponding to a zone of grey dolomite within nearly completely dolomitized WAL mud mounds. At similar depths, in drill hole TC-2638-036, dolomitized WAL mud mounds reveal lower velocities (5700 to 6000 m/s), associated with a brownish dolomite (arrow 4; Fig. 6). High velocities can also be associated with clean zones of the WAL mud mounds, and these are intersected by the drill holes TC-2638-074 and TC-2638-101, displaying velocities of 6000 m/s on both the model and drill holes. These observations strongly suggest that the tomographic velocity model is sensitive to differences in calcite and dolomite textures.

Figure 6: Travel-time tomography model displaying P-wave velocity data alongside petrophysical velocity measurements from the drill core samples (marked by the yellow circles). The purple line outlines the overburden thickness interpreted from the model and drill hole data. Black arrows mark features identified in the model. The coloured top bar represents the bedrock stratigraphy extracted from the geological map.

The velocity model shows interfingered low and moderate velocity zones (arrow 1; Fig. 6), corresponding to 90 meters of altered volcanic sequence of KKR underlain by a dolomitized zone of the LGG intersected by drill hole TC-2638-088. South of 144000mN, the same units are exposed at the surface, with the model revealing a thinner upper low-velocity layer and predominantly high velocities below, indicating a distinct velocity response compared to the northern section of the model. The bedrock geology in this area (Fig. 1) is mapped as KKR.

Younger units (KKS and HBN) are exposed between 146000mN and 144000mN (arrows 1 and 2; Fig. 6). The response in this portion of the tomography exhibits chaotic behaviour with increased lateral velocity variability, interpreted as interfingering

between the volcanic- and limestone-dominated intervals, as supported by drill hole TC-2531-001. The upper 200 meters of this drill hole intersect an interfingered sequence of thickly bedded volcanics, argillaceous limestone, cavities and basaltic intrusions. Laboratory Vp ranges from 4700 to 6200 m/s, aligning well with velocities observed in G11-2531-01 (Fig. 5a), which also intersects KKS, KKR and the limestones of LGG.

# 4.2 Acoustic impedance




By combining the P-wave velocity with the density data from all drill holes, we calculated the acoustic impedance for each data point, as well as an average value for each stratigraphic unit observed in the drill holes, to identify the main acoustic impedance trends (Fig. 7).

Velocity contrasts between individual units within the Limerick Syncline are generally mild. The most notable differences occur between the WAL and ABL (mean velocities of 6000 vs 5500 m/s, respectively, representing a 10% contrast) and between the LGG and the KKR (mean velocities of 6000 vs 5600 m/s, respectively). These velocity differences are also observed in the downhole Vp logs (Fig. 5). Density contrasts are most prominent between the carbonate and volcanic rocks (approximately 2.7 vs 2.9 g/cm<sup>3</sup>, respectively) and the carbonate and mineralisation (2.7 vs 3.0 to 4.2 g/cm<sup>3</sup>, respectively).

The ABL exhibits lower acoustic impedance (15,000 g/cm²s) than the WAL (16,500 g/cm²s). This reflects the lithological composition of the upper ABL, containing significant portions of mud-rich limestone. Where the two units are juxtaposed, an impedance contrast of 1,000 to 1,500 g/cm²s is expected. However, as seen in G11-2638-06 (Fig. 5b), the transition between these units can be gradual, with the top tens of meters of the ABL representing a mudstone-dominant unit with micrite nodules, which is transitional to the WAL (Somerville et al., 1992). As expected, acoustic impedance in the WAL is highly variable due to significant changes in density and velocity (Fig. 7). Most clean samples (Table 4) in the WAL show high acoustic impedance, contrasting with the brecciated and dolomitized samples in the unit (≤ 15,500 g/cm²s).

Figure 7: Laboratory P-wave velocity and density data cross-plot. Data points coloured according to main stratigraphic units. The mean acoustic impedance per unit is outlined in blue. Mineralisation and dykes are distinguished in the plot as MIN and DYK, respectively.

The acoustic impedance in the LGG gradually decreases upwards (e.g. G11-2638-06 in Fig. 5b). However, it is mostly indistinguishable from the WAL. The KKS show very low acoustic impedance (around 13,000 g/cm²s). It is characterised by basaltic lava flows, intrusions, and tuffs, representing thick, massive, homogeneous units (Strogen, 1983). Intense chloriteclay alteration is common in most of the measured KKS samples, typical of extrusive units in both KKS and KKR (Slezak et al., 2023). This may explain the observed low overall acoustic impedance (13,000 g/cm²s) and low velocities (Fig. 5a). Basaltic samples of the KKR show much higher acoustic impedance, mainly between 15,000 and 16,000 g/cm²s, and they are distinctly dense (between 2.75 and 3.00 g/cm³; Fig. 7). The KKR samples are dominated by relatively unaltered fine-grained massive basalt intrusions, and lower values within KKR (13,500 g/cm²s) correspond to volcaniclastic and tuff samples, which are often altered. The HBN shows very similar behaviour to LGG and WAL, and when interfingered with KKR, does not show notable acoustic impedance contrasts.

Massive sulphide mineralisation shows the highest acoustic impedance values ( $\geq 20,000 \text{ g/cm}^2\text{s}$ ) due to the densities associated with pyrite, sphalerite and galena, with an expected impedance contrast of 4,000 to 5,000 g/cm<sup>2</sup>s for mineralised WAL.

#### 4.3 Seismic reprocessing






The velocity analysis provided key information to reconstruct the velocity field in the study area, which informed the new stacking velocity model used in both the stack and post-stack time migration. The new stack of profile LK-11-02 (Fig. 8) features areas with prominent reflectivity. The northern part of the profile contains the Stonepark Zn-Pb prospect (Fig. 1), where several drill holes reach depths of up to 300 meters (CDPs 0-600). This area is characterised by a group of high-amplitude reflections (R1) with variable geometries and lateral continuity between 100 and 300 ms, exhibiting mound-shaped structures with greater impedance contrasts than those observed right below. Towards the south, a sequence of gently to moderately dipping reflections (R2) shows high amplitude and subparallel to mound-shaped geometries. Similar reflections are also observed between CDPs 720 and 1130 and between 1130 and 1335, at 300 to 500 ms (black arrows in Fig. 8).

Figure 8: Two-way time stack section with reflections R1, R2 and R3 highlighted by blue arrows. Black arrows indicate discontinuous features identified in the section.

Laterally continuous reflections (R3) are identified on both sides of the section. The R3 reflections are the deepest continuous reflectors in the profile, below which the reflections become more incoherent. However, there is a significant discontinuity at the central part of the section (between CDPs 720 and 1130), interrupting the R3 package. Notably, these R3 reflectors dip in


#### 5 Discussion

#### 5.1 Impact of reprocessing on seismic imaging of profile LK-11-02

To evaluate the impact of our reprocessing on the imaging of profile LK-11-02, we compared the results of the Kirchhoff post-stack time migration with the Kirchhoff pre-stack time migration of dataset 2 (Fig. 9). Reflections recovered by the pre-stack time migration (Fig. 9a) are mainly concentrated in the shallow subsurface between 100 and 200 ms, where reflections R1 and R2 (mapped in Fig. 8) appear only partially imaged and with chaotic patterns. In contrast, in the post-stack time migration (Fig. 9b), R1 is expressed as mound-shaped reflections and R2 as subparallel reflections. The R3 package is visible in the southern part of the profile (south of CDP 1130) in both migrations, with relatively good continuity. However, the pre-stack migration struggles to resolve reflections on the north side of the section below 200 ms, producing migration smiles.

Figure 9: Two-way time migrated sections from dataset 2 and new reprocessing. a) Kirchhoff pre-stack time migration from dataset 2, and b) new Kirchhoff post-stack time migration. Reflection R1, R2 and R3 are highlighted by blue arrows.

The use of an updated velocity model, combined with stacking before time migration, improved the signal-to-noise ratio by mitigating and reducing the effects of the noisy CMP gathers and complex geology. This resulted in better signal recovery in

both the shallow and deep portions of the section, revealing enhanced continuity and previously unresolved reflections of geological significance. Overall, the reprocessed and newly migrated section sharpens reflector geometries and amplitude variations, making the image more interpretable.

# 350 5.2 Seismic interpretation



Figure 10 shows the interpretation of the main stratigraphic units and structures on a depth-converted profile LK-11-02. The post-stack time-migrated section was depth-converted using a constant velocity of 5500 m/s, with the datum set at 160 m MSL. Drill holes projected onto the section provide geological control, documenting dolomitization, brecciation, mud-rich intervals and dyke intrusions. Black dashed lines outline interpreted major faults F1 and F2, not previously mapped in the region.

- Reflections R3 are attributed to the LLS Group (yellow horizon in Fig. 10b), comprising interbedded cross-bedded sandstones, oolitic and skeletal grainstones and fissile shales with desiccation cracks (Somerville & Jones, 1985; Somerville et al., 2011). This heterolithic sequence could explain the observed high-amplitude, subparallel, continuous reflectors. The upper contact of R3 is interpreted as the transition from mixed siliciclastic-carbonate-evaporite sequences to fully carbonate-dominated ramp sequences of the Ballymartin and Ballysteen Formations (Somerville and Jones, 1985).
- The enhanced continuity of reflections combined with drilling constraints in the northern portion of the section allows recognition of the ABL-WAL contact (dark-blue horizon in Fig. 10b). The overlying WAL exhibits high-amplitude, mound-shaped structures with greater impedance contrasts, corresponding with the observed Vp variability within the WAL (Fig. 7; Table 4).
  - Internal form lines within the WAL (cyan blue in Fig. 10b) outline semi-continuous moderate to high amplitude reflections in mound-shaped geometries with opposing dips, and associated onlap and offlap terminations. These features correlate well with facies transitions observed in drill holes (Fig. 10b), from massive micritic mud mound cores (higher acoustic impedance) to bioclastic-rich flanks containing variable amounts of mud (lower acoustic impedance). Outcrop and drill core evidence across the Irish Midlands show that individual WAL mud mounds can have dimensions of tens to a few hundred meters in thickness (Lees, 1964) with amalgamated mud mound complexes extending laterally for kilometres (Devuyst and Lees, 2001), especially in the Limerick area (Somerville et al., 2011).
  - Distinctive high-amplitude reflection patterns in Fig. 10 are generally associated with igneous rocks. North of CDP 720, shallow, laterally continuous subparallel reflectors at  $\sim$  -100 m (dark pink dashed form line; Fig. 10b) coincide with basaltic intrusions that interfinger with thick WAL mud mound flank facies in TC-2638-101 ( $\geq$  50m), and with flank facies with minor dykes in TC-2638-026 and TC-2638-036 (Fig. 4).
- To the south, high-amplitude, semi-continuous subparallel reflectors (e.g. R2; Fig. 9b) are interpreted as KKR units (dark pink form lines; Fig. 10b). These strong reflectors are often underlain by vertically connected zones of moderate to high amplitude discontinuous reflectivity extending down to ~ -1500 m, and interfingering with chaotic, low amplitude reflections. The KKR comprises abundant basaltic maar-diatremes, lava flows, intrusions, tuffs and volcaniclastic units (Strogen, 1983; Elliott et al., 2015; Slezak et al., 2023). In this context, the high-amplitude mound-shaped reflections between CDPs 1130 and 1335 are interpreted as an intrusive complex, possibly a diatreme pipe with associated intrusives. The surrounding chaotic low-amplitude reflections between -250 and -900 m likely represent volcanic-carbonate interfingering within the LGG and HBN.

The contact between the KKS and the HBN is constrained by the drill hole TC-2531-001 and outlined by the light-blue horizon (Fig. 10b). The acoustic impedance plot (Fig. 7) exhibits low values for the altered volcanic samples, tuffs, and volcaniclastics and high values for basaltic lava flows, reinforcing the contrasts observed in the seismic section.

Figure 10: a) Depth-converted section. b) Seismic interpretation of the main stratigraphic units with drill holes containing geological logging information (units and textures - dolomitization/brecciation, argillaceous intervals and dyke intrusions). Interpretation of faults F1 and F2 are outlined by black dashed lines. Arrows 1 and 2 indicate fault-related features identified in the section. The coloured top bar represents the bedrock stratigraphy extracted from the geological map.


Several interpreted faults crosscut the section. A significant discontinuity interrupts continuity and elevation of the LLS package. This is interpreted as a major south-dipping (segmented) fault (F1) displacing the LLS by approximately 750 m. Additionally, a north-dipping fault (F2) is interpreted to the south of F1, displacing the LLS by approximately 200 m. Reflectors of overlying units are dipping north towards F1, forming a clear normal fault half-graben geometry, with F1 representing a terraced fault system. Several antiformal geometries and domed reflections (arrow 2; Fig. 10b) are observed in the immediate hanging wall and footwall of the fault, as well as in the F1 fault blocks. A zone of more steeply dipping LLS

reflectors is also seen between CDPs 1540 and 1740. These are interpreted as representing inversion geometries, possibly associated with the Variscan Orogeny.

# 5.3 Imaging potential of Zn-Pb deposits and surrounding hydrothermal breccias








According to Salisbury et al. (2000), a sulphide deposit can be imaged using seismic if it meets three conditions: i) a sufficiently high reflection coefficient (R), specifically greater than 0.06 in their study; ii) a diameter greater than the width of the Fresnel zone; and iii) a thickness of at least ¼ wavelength due to seismic vertical resolution.

In our study, although the Zn-Pb mineralisation in the Limerick Syncline has a sufficiently high reflection coefficient and Fresnel zone, it is unlikely to be directly imaged due to its moderate thickness. The mineralisation typically occurs in 10-meter intervals within brecciation zones in the WAL, below the vertical resolution of the seismic data, estimated at approximately 40 meters for a dominant frequency of 40 Hz. However, our acoustic impedance study indicates that the mineralisation exhibits the highest impedance values compared to the main lithologies in the area (Fig. 7). This suggests that if a sphalerite/galenarich zone extended laterally for at least 240 meters (approximate Fresnel zone width) and had a minimum thickness of 36 meters, it could potentially be imaged by the seismic (calculations assume Vp = 5800 m/s, dominant frequency = 40 Hz, and depth = 200 meters).

Hydrothermal dissolution-precipitation breccias are widely associated with Zn-Pb mineralisation in the Irish Midlands (Hitzman, 1999; Wilkinson et al., 2011; Güven et al., 2023) and are also recognised at Stonepark and Pallas Green (Elliott et al., 2019; Blaney and Coffey, 2023). At Stonepark, brecciation zones within the WAL display lower acoustic impedance values compared to unaltered carbonate samples. Drill holes intersect WAL breccia zones between CDPs 100 and 515, at depths from -100 to -300 m (Fig. 10b), coinciding with a chaotic, low-amplitude seismic facies. This contrasts with the seismic characteristics of WAL mud mounds (Fig. 10b) and dolomitization, whose seismic responses closely resembles that of the mud mounds. The distinctive seismic character of these brecciated zones, as confirmed by drilling, therefore highlights their potential as indirect seismic proxies for potential Zn-Pb mineralisation. Our work suggests that directly incorporating breccia identification into seismic interpretation may sharpen drill targeting. To further evaluate its effectiveness, we will apply this reprocessing workflow to the other five Stonepark and Pallas Green seismic profiles in the future.

The occurrence of breccias and mineralisation at Stonepark and Pallas Green also ties into the structural framework (Blaney and Coffey, 2023). Given the proximity of the Stonepark deposit to fault F1, the structure could have acted as a master conduit for metal-bearing fluids within the overall mineral system (Torremans et al., 2018; Walsh et al., 2018), penetrating deeply into the Lower Palaeozoic basement rocks, the source of metals for mineral systems in the Irish Orefield (Johnston et al., 1999; Wilkinson, 2023). In addition, the presence of significant displacement on F2 in the southern side of the section (Fig. 10b) suggests it could also have served as a hydrothermal fluid pathway, further opening exploration search space on the opposite flank of the Limerick Syncline.

## 5.4 Implications for the architecture of the Limerick Syncline

The Limerick Syncline forms a regional ovoid-shaped map pattern whose structural evolution remained poorly understood. The Stonepark and Pallas Green deposits, located in the northwest portion of the Syncline, are crosscut by a series of small throw, north-northwest-striking faults that appear to locally control the distribution of mineralisation (Blaney and Coffey, 2023) but which lack demonstrable growth sequences. However, the main basin-controlling and potentially ore-controlling normal faults remained unidentified in the study area (Blaney and Redmond, 2015). Our velocity analysis and seismic reprocessing approach have clearly resolved and delineated with confidence a major south-dipping fault (F1) and a north-dipping fault (F2) to the south of Stonepark and Pallas Green.

Considerable thickness variations are observed along profile LK-11-02 (Fig. 10). The Tournaisian units, LLS and ABL, appear to show no significant thickness change, consistent with their depositional setting as a shallow-marine, tidally influenced environment on an inner ramp, grading upward into more mud-dominated shelf/ramp conditions that precede WAL buildups (Somerville and Jones, 1985; Somerville et al., 1992). They are therefore interpreted to be largely deposited in a pre-rift setting. In contrast, the WAL, LGG, and KKR volcanic sequence thicken in the hanging wall of F1, implying syn-depositional fault growth during the late Tournaisian to early Viséan. Together, F1 and F2 form a half-graben geometry, which has not previously been recognised or described for the Limerick Basin. These fault growth timings are consistent with those observed in the northern and central Midlands of Ireland, where faults are active from the middle Tournaisian, significantly influencing basin development from the Viséan onward (Johnston et al., 1996; Hitzman, 1999; de Morton et al., 2015; Ashton et al., 2018). There, coincident mineralisation was caused by hydrothermal fluid flow along the developing fault systems (Torremans et al., 2018; Güven et al., 2023).

The south-dipping fault (F1) exhibits two fault strands bifurcating from a common point at depth, suggesting a segmented normal fault geometry, with gentle rotation of the reflectors (arrow 1, Fig. 10b) between the two interpreted segments (CDPs 515 to 930). Evidence of this fault zone is also visible in the tomography model, where increased lateral velocity variations occur near drill hole TC-2638-088 (Fig. 6). This structure is similar in nature to normal fault arrays observed elsewhere in the Irish Midlands (Kyne et al., 2019) and, therefore, strongly indicates that large-scale relay ramp geometries are present in the Limerick Syncline (Walsh et al., 2018), increasing the likelihood that further significant normal faults are to be identified in the region. In the hanging wall and footwall of F1, domed reflections (arrow 2, Fig. 10b) are interpreted as representing inversion geometries (Bonini et al., 2012). These are likely associated with the Variscan Orogeny, which led to regional north-south compression and consequent basin inversion, reactivating normal faults and producing reverse faulting (Meere, 1995; Fusciardi et al., 2003; Kyne et al., 2019). Similar domed geometries are also observed on the south side of the profile, dipping towards fault F2. This observation of significant inversion tectonics on km-scale normal faults, and its confluence at the edge of the present-day Limerick Syncline, explains why these Mississippian normal faults have hitherto not been recognised, since current net displacements on F1 are mostly reverse at present-day erosion levels.

In addition to identifying faults, the new processing has also enhanced our understanding of the distribution of the volcanic sequence along the seismic profile, which shows some association with faulting. The interpreted basaltic intrusions, located north of CDP 720 (Fig. 10b), are interrupted by the segmented fault (F1). This high-amplitude reflector package reveals an increased thickness between CDPs 515 and 720 compared to the geometry observed north of CDP 515, suggesting coeval faulting and volcanism in the region, both culminating during the Viséan. High-amplitude and laterally continuous subparallel shallow reflectors, such as those linked to the basaltic intrusions, are also noted south of CDP 1740 at a depth of -100 m, intersected by fault (F2). This indicates that the southern side of the profile displays a geological setting very similar to that of the northern side, where the Stonepark deposit is situated.

# **6 Conclusions**

This study demonstrates how selecting an appropriately tailored processing workflow, guided by geological and petrophysical data, can substantially enhance seismic data interpretation. Incorporating additional datasets, such as drilling information, downhole and laboratory P-wave velocity measurements, and travel-time tomography into the stacking velocity model and post-stack migration has improved subsurface imaging on profile LK-11-02, sharpening continuity and fidelity of reflectors. The velocity and acoustic impedance analysis provided valuable insights into the imaging potential of unit boundaries, brecciation zones and mineralisation.

The reprocessed seismic section resolves key stratigraphic and facies architecture previously obscured, including the ABL-WAL contact, internal WAL mud mound form lines, and the lateral interfingering of volcanic (KKR/KKS) and carbonate (LGG, HBN) units. Importantly, we identify two previously unmapped, basin-scale structures: a south-dipping segmented normal fault (F1) and a north-dipping fault (F2) that together bound a syn-rift (half-)graben geometry. Their geometry and associated reflection patterns are consistent with late Tournaisian to early Viséan syn-depositional faulting during Mississippian rifting, coeval with emplacement of the Limerick Igneous Suite, and followed by Variscan inversion tectonics, which provide net zero displacements at the present-day surface. Given its position near Stonepark, the south-dipping F1 is a plausible hydrothermal fluid conduit for the mineral system. Furthermore, this study highlights that the southern side of the Limerick Syncline, where our new imaging workflow reveals a major north-dipping fault, F2, warrants follow-up investigation.

From the acoustic impedance analysis, sulphide mineralisation exhibits high reflection coefficients. However, the typical 10 m-thick intervals are below the quarter-wavelength vertical resolution necessary to be directly imaged at ~ 200m depth with the available seismic data. Consequently, direct seismic imaging of similar discrete sulphide bodies as known in the Limerick Syncline is unlikely. However, WAL breccia corridors, often surrounding Zn-Pb mineralisation, display lower acoustic impedance and are mapped as chaotic, low-amplitude seismic zones, confirmed by drill hole intersections. Where these corridors are laterally extensive, they may provide a practical indirect seismic proxy for Zn-Pb targeting.

Lastly, the workflow explored in this study presents a cost-effective and transferable approach to mineral exploration. By re-evaluating seismic data with added petrophysics, valuable new geological insights can be extracted, offering an alternative to, or supporting the decision to, acquire new seismic data. Moreover, this approach enables the recovery of additional geological information from datasets that required significant financial and operational effort to acquire, thus contributing to the revalorisation of existing resources and maximising their exploratory potential.

#### Data availability





Seismic data underpinning this work can be requested through the Geoscience Regulation Office (GSRO) of the Department of the Environment, Climate and Communications of the Government of Ireland. The petrophysics data generated in this work can be obtained upon request from the authors. Measurement metadata reports per drill hole are published in a research repository (Zenodo). Other complementary data can be obtained from the authors upon request.

#### **Author contributions**

VS: conceptualisation, petrophysical data acquisition, seismic, tomography and petrophysical data processing, data interpretation and integration, writing (original draft preparation) and editing; AM: conceptualisation, project management, supervision, funding acquisition, reviewing; KT: conceptualisation, data interpretation and integration, supervision, reviewing; JA: seismic processing methodology, supervision, reviewing; DM: travel-time tomography methodology, supervision, reviewing; RB: seismic processing methodology, reviewing.

# Competing interests


The authors declare that they have no conflict of interest.

# Acknowledgements

We greatly thank Group Eleven Resources, particularly Mark Holdstock and Sean Walsh, for providing geological logging data and drill core samples used in this study for laboratory petrophysical measurements, as well as for valuable geological discussions. We also acknowledge Robertson Geo for downhole petrophysics data acquisition. We are grateful to Glenn Morgan, Earth Signal Processing, and VelSeis Processing for providing the legacy seismic datasets used in the new processing discussed in this study. Additionally, we appreciate AspenTech, RadExPro and SLB for granting academic licenses for their software, which is used as the main tool in seismic processing. Special thanks to Philip Rieger and Eoin Dunlevy for their invaluable support in geological discussions. Finally, we extend our gratitude to the Geological Survey Ireland for hosting the UCD Petrophysics Laboratory from 2021 to 2023 and to research assistants Conor Farrell, Eoin Byrne and Gabriel Cavalcanti for their valuable support in the petrophysics laboratory.

# Financial support





This research was partly supported by a UCD Ad Astra Studentship and partly supported by Science Foundation Ireland, now Research Ireland, projects 13/RC/2092\_P2 and 16/RP/3849. JA and DM acknowledge funding from the VECTOR project; this research has received funding from the European Union's Horizon Europe research and innovation programme under grant agreement no.101058483 and from UK Research and Innovation. RB acknowledges funding from Grant PID2024-159984NB-I00 (DESTINY) funded by MCIN/AEI. This is a B-CSI publication (2021 SGR 00429).

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
