# Peer review of "Petrophysics-guided velocity analysis and seismic data reprocessing to improve mineral exploration targeting in the Irish Zn-Pb Orefield"

_EGUsphere, 2025_

## Author Response (AR1)

**Referee: Samuel Zappalá**

Dear Samuel,

Thank you very much for taking the time to review our manuscript. Your comments were very helpful in preparing a clearer and more concise revised version of the study. Below, we provide our responses to each of your comments and questions. We hope that we have addressed them satisfactorily.

**Related questions:**

(1) the petrophysical velocity analysis from the cores matches only partly the sonic logs with a big number of outliers (see figure 4b after 300 m of depth or the low measurements coherency in figure 4a), you justify this behavior in the text with understandable reasons, but how do you handle the problem? Or you just ignore it?

**Response:** The comparison between laboratory P-wave velocities from core samples and sonic log data is intended primarily to validate the laboratory measurements and check whether the general velocity trends are coherent with the in situ downhole measurements. While we do observe differences, they are not explicitly addressed in the analysis because they result from factors such as instrumentation differences, anisotropy, and other geological effects. Our aim is not to correct or fully reconcile these differences, but rather to confirm that the trends seen in the laboratory velocities are consistent with the sonic logs and can be used reliably for the velocity analysis.

(2) have you tested different starting model for the tomography (simpler or more complex)? Did you remove the petrophysical constrains to evaluate if they are really influencing the final tomography results? did you evaluate how much it is sensible to the inaccuracies in the velocities from the core samples?

**Response:** Pstomo\_eq is a highly robust software in terms of its convergence towards reliable final velocity models, allowing the use of relatively simple initial velocity models. The initial inversions were performed using basic models (1D) with smooth velocity gradients and without introducing lateral velocity variations. These early inversions were primarily aimed at assessing the quality of the picks and identifying potential errors in the acquisition geometry. Even these preliminary inversions yielded excellent convergence, with a significant reduction in misfit (i.e. one order of magnitude).

Based on the results of the initial inversions and supported by petrophysical data, smooth lateral velocity variations and variable velocity gradients with depth were introduced in the initial model to reflect trends observed in both datasets. The aim was not to use a highly detailed initial model, but rather to reflect these trends without compromising the independence of the inversion from the initial model. These updated models facilitated faster inversion processes, allowing us to reduce the size of the inversion grid, which resulted in an improvement in the resolution of the final velocity model.

(3) why did you not use a smoothed version of the tomography model for stacking the data (at least for the shallow depth) instead than computing another velocity model with semblance analysis?

**Response:** The stacking velocity is computed for NMO (Normal Moveout) correction, which does not necessarily reflect the true subsurface velocity field. Stacking velocity is an effective velocity that best flattens reflection events in a common midpoint (CMP) gather, assuming horizontally layered, planeparallel interfaces. However, this assumption often does not hold in real geological settings, where layers may be dipping, curved, or heterogeneous.

In contrast, interval velocity represents the actual seismic velocity within a specific layer, derived from Dix's equation using stacking velocities and two-way travel times. Therefore, discrepancies between

stacking and interval velocities arise due to simplifications in the NMO model, lateral velocity variations, and vertical heterogeneities.

There are also logistical challenges involved in transforming the tomographic inversion into a usable velocity model for the entire seismic section. This includes resampling and extending the model, which is complicated by the fact that the CDP binning does not align precisely with the tomographic section, as the latter is computed in 3D. As a result, reconciling both domains requires careful adjustment. While we considered using a smoothed version of the tomography model for migration, it was ultimately not applied due to these challenges and the project time constraints.

In any case, the tomographic inversion was conducted to validate the petrophysical measurements and to support the velocity model building to be used in reflection seismic processing. Although the type of analysis suggested by the reviewer could be of interest, it falls outside the scope of the objectives pursued with seismic tomography. The availability of some picks from legacy data enabled the use of an additional methodology to complement the main goals of this study, which are primarily focused on petrophysical characterisation and the reprocessing of reflection seismic data.

**(4) did you try to use constant velocity stack analysis instead than semblance? It is usually effective in hardrock domain but I am not sure what will be better in this mixed case.**

**Response:** Given the highly variable lithology of the study area, carbonate-volcanic interfingering and intense alteration of the carbonate rocks, the constant velocity stack analysis was not suitable for coherently imaging reflections across the section. The semblance analysis provided a more reliable basis for the velocity model.

**(5) it is not clear what velocity model you are using for the migration, is it exactly the same one that you use for the stacking?**

**Response:** The model used for migration was the smoothed stacking velocity. We have clarified this in the text as follows:

Line 222-223: 'The final step in the processing sequence was a Kirchhoff post-stack time migration, using 100% of the smoothed stacking velocity.'

You should compare the migrated section resulting from your proposed approach with the one from the newest dataset (dataset 2) and not with the old one (dataset 1) for 2 main reasons: (1) Dataset 2 supposedly shows better results since it has a more recent processing and (2) dataset 2 is the starting data you used for your sections. It would be interesting and useful to compare also the velocity model used from the previous study that you use for comparison.

What you are comparing now (Figures 9a and b) just shows that two different processing returns two different results, nothing new in seismic. The differences between the two processing flows are too many to be able to state that they are due to the proposed velocity approach. They could as well be due to the pre-stack applied processing before your work, or to the different applied migration method and not related at all to the velocity analysis that you performed. So, this point (that I would say is the core of the manuscript) needs to be strengthened.

**Response:** We appreciate the reviewer's comments and the opportunity to clarify this aspect of our work. The comparison with dataset 2 is indeed a logical suggestion; however, this dataset is confidential. Our decision to compare with dataset 1 was derived from this limitation. Unfortunately, neither the original pre-stack data nor the velocity model used in dataset 1 is available to us. As a result, it was technically infeasible to apply our proposed workflow, which includes velocity model refinement and migration steps, to this dataset. Dataset 2 represents the only case where partially processed data and sufficient metadata were accessible to allow reprocessing and integration of petrophysical constraints.

We have now been authorised to publish the pre-stack time migration section from dataset 2, and the manuscript has been revised accordingly. As we now compare our results directly with dataset 2, we have also simplified the 'Seismic data' section, keeping only the workflow description of dataset 2.

We believe this situation reflects a common challenge when dealing with legacy data in real-world mineral exploration or mining contexts: incomplete datasets, missing processing parameters, and insufficient documentation of prior workflows. Our study aims to illustrate a practical example of how existing information can be re-evaluated and enhanced under such conditions, by making the most of available data and integrating independent sources such as petrophysical measurements. We emphasised this rationale more clearly in the revised manuscript.

In the title you say "petrophysics-guided reprocessing" but only the velocity analysis is partially guided from the petrophysical data, everything else is a basic seismic processing flow.

**Response:** Thank you for the valuable comment. The reviewer is correct: the petrophysical data mainly guided the velocity analysis, while the rest of the processing followed standard workflows. We revised the title to reflect this more accurately: "Petrophysics-guided velocity analysis and seismic data reprocessing to improve mineral exploration targeting in the Irish Zn-Pb Orefield"

Figure 6 needs some work to be clearer. In general, the figure is too small and is not possible to see what you describe in the text. (1) Boreholes are too thin and they have no frame making difficult to identify them when they overlap with the tomography section. (2) Some of them is plotted behind the tomography and overlap is not visible at all making impossible the comparison. (3) Do you need to plot the whole depth of the boreholes' velocities? There is nothing to compare them below the tomography model and they do not give any useful information. (4) get rid of the single rays propagating deeper in the model, they are very confusing and not trustable. Once you clean up the figure you can add some zoom in window to highlight the details you are describing in the text. Use this figure to show the quality of your velocity model and how it benefitted from the petrophysics data.

**Response:** Thank you for this comment. This figure has been modified.

In chapter 4.2 It is not clear how you measure the acoustic impedance, you talk in the text about acoustic impedance related to the different formations but how do you calculate it? Is it computed considering the interfaces for each borehole? Is it an average of all density/velocity values for those formations? Is it between each measured point? Explain it more in detail.

**Response:** We have clarified this in the text.

Line 293-295: 'By combining the P-wave velocity with the density data from all drill holes, we calculated the acoustic impedance for each data point, as well as an average value for each stratigraphic unit observed in the drill holes, to identify the main acoustic impedance trends.'

Technical corrections

Line 43 – It should be (Eaton et al., 2010).

**Response:** It is originally from Eaton et al. (2003).

Line 48 – It should be (Wilkinson et al., 2015).

**Response:** The citation has been modified (line 49).

Line 52 – what do you mean by "1000 line-km"?

**Response:** This sentence has been modified to '1,000 km of 2D reflection seismic' (line 53).

Figure 1 – Very nice figure, just some comment. a) and b) should be bigger and bold, but you can actually remove them and just refer to Ireland map as an inset. You may also add a thin black frame to the all figure, but this is just an aesthetical thing. Modify the caption as "...showing the seismic lines, the drill holes...". Highlight the seismic line that is used in the manuscript (i.e. write in red LK 11-02 on the figure).

**Response:** The figure and caption have been modified according to the reviewer comments. Profile LK-11-02 has been outlined in red on the map.

Line 86 – It should be (Chew et al., 2009).

**Response:** The citation has been modified (line 86).

Figure 2 – It is not clear what is the transparent pink that cut through the other formations. Write it in the caption.

Along the manuscript use or the full names for the geological formations or their abbreviations and be consistent both on the text and in the figures/tables, now you are switching between them and this makes it difficult to follow it. If you want decide to use the abbreviations keep their full name only the first time (in the geological chapter).

**Response:** The figure, caption and abbreviations have been modified in the text.

Instead than table 1, a figure showing the boreholes lithological sections will add useful information and it will be easier to follow.

**Response:** The figure has been added in the revised manuscript as Figure 4.

Line 145 – move the sonic log information on chapter 3.1

**Response:** We thank the reviewer for this comment and would appreciate further clarification to ensure we address it appropriately.

Line 158 – "Robertson Geo Mining & Minerals" should be a reference? I cannot see it in the reference list.

**Response:** This has been added to the reference list (line 625).

Line 177 – seismic data – give more information regarding the acquisition. How heavy the seismic vibrators are? What kind of sweep are they generating? What frequency range is used? What sampling rate is used? What maximum and average offset is recorded? How is the raw data quality?

**Response:** This has been added in the text (lines 184-187).

Line 196 and table 2 - What type of velocity analysis has been performed in dataset 1 and 2?

**Response:** This has been added to Table 2 (line 206).

Figure 4 – make the figure bigger or increase the font size of the text in the figure, it is too small to read. You are again using different names for the stratigraphic units, I would suggest to use the abbreviations.

**Response:** The figure has been updated according to the reviewer's suggestions and is now Figure 5.

Line 257 – it should be (Lees and Miller, 1995).

**Response:** This has been removed from the text.

Line 276 – It should be (Sleeman et al., 1999).

**Response:** This has been removed from the text.

Line 355 - I do not see these reflections observed between the CDPs 720 and 1130 and between 1130 and 1335, around 300 to 500 ms that you refer to. There is some discontinue reflection but hard to define them.

**Response:** Arrows have been added in Figure 8 to facilitate the identification of these features.

Figure 8 – Is it migrated or unmigrated? I cannot see many differences from the one in figure 9b.

**Response:** This is the unmigrated stack section. The two images appear similar because most of the reflections exhibit moderate dips, and the migration was performed with a smoothed stacking velocity model. As a result, the migration mainly enhances amplitudes and continuity rather than repositioning events.

Line 356 – I am not so confident about the R3 reflection on the north. It is clearly visible the northern part but it is not easy to follow it after CDP 250. You should move the second arrow closer.

**Response:** We thank the reviewer for this comment. The R3 reflections are disrupted by faults on the north side of the profile, which makes it difficult to track the reflection consistently.

Line 368 – You cannot refer to figure 9c before you refer to figure 9a (or to figure 9 in general).

**Response:** This section has been revised. Section 5.1 is now titled 'Impact of reprocessing on seismic imaging of profile LK-11-02', and 5.2 is 'Seismic interpretation', addressing the figure reference issue. A new figure has also been added, presenting a depth-converted section along with interpretation (Fig. 10).

Figure 9 – In the legend you write the name of the formations to indicate what is a horizon. Write if it is Top or Bottom of the formation.

**Response:** The figure and legend have been modified accordingly and are now Figure 10.

Line 404 – I do not see any light green horizon in the figure, maybe the light blue.

**Response:** This has been modified (line 380).

Line 408 – There is any information about these faults in literature? If yes it will be useful to reference them.

**Response:** These faults have not been previously mapped. No major structure has been detected or intersected from drilling in the Limerick Syncline to date (Blaney and Redmond, 2015; Blaney and Coffey, 2023).

Line 439 – Which dominant frequency are you considering in your calculations?

**Response:** The dominant frequency is 40 Hz. The information has been added to the text (rows 399-402).

State clearly the uncertainty of your interpretation, especially for the reflections that are not intersected by the boreholes and for the faults.

Line 466 – it should be (Fusciardi et al., 2003).

**Response:** This has been modified (line 446).

Line 572 – de Morton 2015 (1) is not in the text.

**Response:** This has been properly referenced.

Line 646 – You should distinguish the two Slezak 2023 references according to the authors guidelines.

**Response:** This has been properly formatted.

**Referee: Anonymous**

We thank the reviewer for the time and effort dedicated to reviewing our manuscript. The detailed comments have helped us improve the clarity and conciseness of the study. Below, we provide our responses to each of the comments and questions. We hope that we have addressed them satisfactorily.

**Comments:**

My first comment is related to the term "legacy data". I have failed to understand the reasoning behind. To me, it is simply a 2D seismic data acquired in 2011 (so relatively recent), which the author(s) have used to obtain a better subsurface image. Please provide appropriate justification; otherwise, I strongly advise removing this term from the article to avoid giving a wrong impression to the readers. Also, it is quite strange to read terms like legacy sonic log, legacy first-break pics, and so on throughout the article [see Row 115-116]. I advise removing such terminology once the dataset is properly introduced [section 3].

**Response:** Thank you for your comment. We originally used the term 'legacy'to simplify references to previous processing versions. However, we understand your concern regarding the terminology and have removed it from the text. The same adjustment has been applied to the references to the sonic log and first-break picks.

The second comment is related to the two datasets described in section 3. I have failed to understand how the first dataset 1 processed by Velseis Processing Ltd is relevant to this study as the authors have chosen dataset 2 for their purpose. It is natural to obtain two different results if two significantly different processing workflows are followed [Table 2]. I advise limiting the data description to dataset 2 only as this was used as input for further improvement. Regarding the use of first breaks from dataset 1, it can be simply mentioned that they were available and further corrected, and tomography was performed to obtain a P-wave velocity model.

**Response:** Previously, we described dataset 1 as our results were compared with its phase-shift time migration (Fig. 9). We have now been authorised to publish the pre-stack time migration section from dataset 2, and the manuscript has been revised accordingly.

Since we now compare our results directly with dataset 2, the 'Seismic data' section has been simplified to include only the workflow description of dataset 2, as suggested.

This is related to the choice of results used for the comparison. As dataset 2 is our input dataset, I will expect to see all the comparisons concerning to it [Row 195-196]. Instead, following the workflow shown in Figure 3, and the results shown in Figure 9 – PoSTM result from dataset 1, (a) is compared with its improved version in (b). Shouldn't it be that the Pre-STM result available from ESP [Row 182-183] is to be compared with the new processed results? Or if the reason given in Row 213-215 is followed, PoSTM results can be easily produced for dataset 2 and can also be compared. But given the complex geological setting of the Limerick syncline, PreSTM is a more natural choice and should produce a better image – probably that is the reason ESP performed PreSTM instead of PoSTM by Velseis.

**Response:** We appreciate the reviewer's comments and the opportunity to clarify this aspect of our work. The comparison with dataset 2 is indeed a logical suggestion. However, this dataset is confidential. Our decision to compare with dataset 1 was derived from this limitation. Unfortunately,

neither the original pre-stack data nor the velocity model used in dataset 1 is available to us. As a result, it was technically infeasible to apply our proposed workflow, which includes velocity model refinement and migration steps, to this dataset. Dataset 2 represents the only case where partially processed data and sufficient metadata were accessible to allow reprocessing and integration of petrophysical constraints. We have now been authorised to publish the pre-stack time migration section from dataset 2, and the manuscript has been revised accordingly.

We believe this situation reflects a common challenge when dealing with legacy data in real-world mineral exploration or mining contexts: incomplete datasets, missing processing parameters, and insufficient documentation of prior workflows. Our study aims to illustrate a practical example of how existing information can be re-evaluated and enhanced under such conditions, by making the most of available data and integrating independent sources such as petrophysical measurements. We emphasised this rationale more clearly in the revised manuscript.

Regarding the decision to perform a post-stack migration, we have clarified this in the revised text (rows 225-228). The CMP gathers in the dataset are noisy, and stacking before migration improved the S/N ratio, providing a clearer seismic image. In addition, the post-stack approach helped suppress migration smiles and enhanced reflector continuity, which provided more interpretable results than those obtained with pre-stack migration.

Combining the two comments above, it is my understanding that the semblance analysis was performed on the CDP gathers of dataset 2 because it has a better signal-to-noise ratio as compared to dataset 1 [Row 190], and PoSTM has produced better imaging (dataset 1) than PreSTM (dataset 2). If this is the case, it is unclear in the article; kindly make this distinction clearer in the article on the datasets and the obtained results. But again, to my reservation, following Table 2, I have an impression that dataset 2 should have provided better results given the use of refraction analysis, two passes of velocity analysis, surface-consistent statics, residual scaling, etc. I suggest adding a figure showing a comparison of raw data with the previous processing by Velseis/ESP, or both at various stages. And then compare it with the new results. I understand that not everything can be shown, but at least the main steps can be shown as this is the core of the article i.e. seismic processing.

**Response:** We have clarified the workflow in the revised manuscript within the sections 'Data and methods', 'Seismic imaging', and 'Impact of reprocessing on seismic imaging of profile LK-11-02'. Regarding the comparison with dataset 2, we kindly note that this point was addressed in our previous response, and we hope that our clarification is satisfactory.

I also have an unwise impression that the stack section shown in Fig. 8 is the same as Kirchhoff PoSTM in Fig. 9b. We expect the events to move up-dip after the migration; kindly cross-verify. The same is for Fig. 9a, looking at the migration smileys – it appears to be from PreSTM although it was not performed on dataset 1. Kindly cross-verify if there is no mix-up during the production of the figures.

**Response:** Figure 8 presents the stack section, whereas Figure 9b shows the post-stack migration, which was generated using the stack section as input. The two images appear similar because most of the reflections exhibit moderate dips, and the migration was performed with a smoothed stacking velocity model. As a result, the migration mainly enhances amplitudes and continuity rather than repositioning events.

I did not understand the reason behind the time-converted drill holes. How was it done? Shouldn't the interpretation of the final sections be converted to depth? Without it, how do we compare it with the information provided in section 2, in particular, to the thickness of the different formations shown in Fig. 2?

**Response:** We appreciate the reviewer's comment. Following your suggestion, we have included the interpretation of the depth-converted section in the revised manuscript (Fig. 10). This has been integrated with the projection of the drill holes onto the profile to better constrain the interpretation. The seismic datum, originally set at mean sea level, was adjusted to a flat datum above the topography (160 m MSL) to provide a consistent reference for the interpretation. As noted in the text, the drill holes are not located directly on the seismic profile, but are situated a few hundred meters away from the seismic line (Table 1).

Right now, it is extremely difficult to compare the seismic results (time-sections) with the direct information, i.e. drillholes and traveltime tomography results (both are in depth-domain), also not defined from a common depth/elevation reference point. There are starting and ending depths provided for the boreholes in Table 1 but there is no information from which elevation/depth level they are defined. They can be either be defined w.r.t. the mean sea level or the seismic reference datum. I advise synchronising all the results with a chosen reference level such that uniformity can be achieved for all results.

**Response:** We thank the reviewer for this comment and the opportunity to provide clarification. We have corrected this inconsistency in the manuscript by adding a depth-converted section (Fig. 10), clarifying the reference datum in the FATT (surface topography; rows 256-257), and including a fence diagram (Fig. 4) that uses topography as a common reference level for the lithological sections of the drill holes.

Table 1 provides little information on the drill holes. I request that lithological sections be added for all boreholes defined from a common reference level. If this reference level is chosen differently from the seismic reference datum, it is to be clearly mentioned.

**Response:** We thank the reviewer for this suggestion. In the revised manuscript, we have added a new figure (Fig. 4) showing the lithological sections of the drill holes, as noted in the previous response.

Currently, it is unclear how the reprocessing is petrophysics-guided. Using petrophysics data for qualitative analysis of the obtained velocity model (from FATT in our case) is pretty standard, or doing seismic-to-well-tie per se. Unless a joint inversion of petrophysical data with FATT is performed or it is used as a constraint, the term "petrophysics-guided" is redundant and I advise removing it from the title, unless a proper justification is provided regarding it.

**Response:** Smooth lateral velocity variations and variable velocity gradients with depth, derived from petrophysical velocities and formation depths, were incorporated into the initial model used for tomographic inversion. The aim was not to use a highly detailed initial model, but rather to reflect trends without compromising the independence of the inversion from the initial model.

We agree with the reviewer's observation: the petrophysical data informed the velocity analysis, while the rest of the processing followed standard workflows. In response, we revised the manuscript title to more accurately reflect this approach: "Petrophysics-guided velocity analysis and seismic data reprocessing to improve mineral exploration targeting in the Irish Zn-Pb Orefield".

As FATT is one of the most important components of the reprocessing, there is hardly any technical detail provided in section 3.3 such as offset information, grid samplings for forward modelling & inversion, picking uncertainty, etc. This section needs to be significantly expanded. Also, how the petrophysical data was used in building the velocity model is not clear.

**Response:** We thank the reviewer for this comment. This information has been added to the text of the section 'Travel-time tomography'.

At this stage, it is difficult to estimate if the improvement in the seismic imaging is mainly due to the velocity model or data processing. Please provide clear evidence for it as this is the core theme of the article.

**Response:** The observed improvements are a result of the updated velocity model and the use of post-stack migration, taking into account the geological context and data quality. We hope that the revisions in the manuscript now provide a clear and satisfactory explanation of our objectives and workflow.

The bibliography section needs to be revised as per the Solid Earth's guidelines.

**Response:** The revised manuscript includes the corrected bibliography section following SE's guidelines.

Minor comments:

1. Row 14-15: Add some information on what Irish-type deposits are like.

**Response:** This has been added in the text (row 14).

2. Row 15: Add a sentence or two describing the general geological setting and then jump to interfinering (specific to the current study). Currently, there is a lack of flow between the two sentences.

**Response:** This has been modified in the text.

3. Row 19: Remove "LK-11-02".

**Response:** This has been removed from the text.

4. Row 41: "...mineral exploration tool." (Add citation)

**Response:** The citation has been added in the text (row 42).

5. Row 46: change to 'Irish-type'

**Response:** This has been modified in the text (row 47).

6. Row 51: change to 'seismic reflection survey'

**Response:** This has been modified in the text (row 52).

7. Row 54-55: Locations of Stonepark and Pallas deposits are not marked in Figure 1

**Response:** We thank the reviewer for the comment. The locations have been added to Figure 1.

8. Row 55: Remove 'the study area of this contribution'

**Response:** This has been removed from the text.

9. Row 80-81: Remove 'Using the resulting improved image'

**Response:** This has been removed, and the text has been modified (row 78-82).

10. Row 116, 118, 119: Remove the term 'legacy' here and in all the forthcoming sentences

**Response:** The term has been removed from the text.

11. Row 123: Remove 'associated'

**Response:** This has been removed from the text.

12. Row 127: Add reason here for why only 2?

**Response:** There have been fewer drill holes drilled into the centre of the Syncline, logically given the large target depths to the target horizon for mineralisation (base of WAL). Only two lie close to the seismic line. These drill holes are of excellent quality, and we have therefore maximised the available data in this study.

13. Row 158: Add citation or remove the reference. It is pretty common knowledge.

**Response:** A citation has been added (row 161).

14. Row 166: Define LSQR

**Response:** This has been modified (row 172).

15. Row 177: Keep only 'Seismic Data'

**Response:** This has been modified (row 183).

16. Row 179-184: Is there an open-access processing report available for these datasets? Please cite here?

**Response:** Dataset 1 with the report can be obtained upon request at the Geoscience Regulation Office of the Department of Climate, Environment, and Communications of the Government of Ireland, as per row 190. Dataset 2 is confidential data belonging to Group Eleven Resources as per row 193. Precisely for this reason, an outline of the original processing workflow of dataset 2 is given.

17. Row 186: I suggest removing the term 'standard', or else define what standard means in this context.

**Response:** This has been removed from the text.

18. Row 189: Remove 'historic' and 'slightly' – it is hugely different

**Response:** The terms have been removed, and the text modified (rows 201-205).

19. Row 195-196: Change CDP to CMP for common abbreviation. You can put the sequence number for ease to know exactly at which step the data is taken, at 16 from dataset 2?

**Response:** This has been modified (row 204).

20. Row 204-205: 'guided by' – this sentence is very loosely defined. At this stage, it is not very clear how exactly petrophysics and tomography were combined. This part needs a clear description. Showcase clearly how your approach is different from other regular methods.

**Response:** The use of petrophysical data in the FATT is now explained more clearly in rows 175-176. The integration of the combined velocity information into the reprocessing workflow is described in rows 215-218.

21. Row 231-232: "downhole petrophysical is rarely available in mineral exploration" – I disagree with this sentence given the fact that still until now, most of the seismic studies for mineral exploration are brownfield studies where the study of the physical rock properties is the first step towards it. Contrary, it is rare to see that petrophysical properties are not available.

**Response:** We appreciate the reviewer's comment. Unfortunately, petrophysical datasets are still not commonly available in the Irish mining industry. We have removed the sentence from the text.

22. Row 222-317: This section is extremely long and very hard to follow. I recommend rewriting with a focus on the main scope of the paper. Extreme details had been provided related to the geology which are indeed very important but do not hold significant importance for this article. It can be part of another paper dedicated to the interpretation of the Limerick Syncline based on the new results.

**Response:** We thank the reviewer for this comment. The section has been shortened, and details regarding the petrophysical characteristics of the carbonate textures have been removed or simplified (rows 233-291).

23. Row 285: "far offsets were included" – please provide some details on this here or in section 3.3

**Response:** This has been added to section 3.3 (rows 164-168).

24. Row 307: "shows interfingered low and moderate velocity zones" – I recommend using arrows or similar objects when describing any events for the ease and do not leave the reader to its imagination. Please consider a similar approach while describing other events in the article.

**Response:** This has been modified in the revised manuscript.

25. Row 319-320: It is unclear whether the data shown is produced from a single borehole or all nearby boreholes. Kindly mention it!

**Response:** The calculations were performed using samples from all the nearby drill holes; this has been clarified in rows 293-295.

26. Row 347: Replace "informed" with "provided"

**Response:** We believe that 'informed' is the most appropriate term to describe the role of the velocity analysis in the stacking velocity model.

27. Row 355: Doesn't "CDP 720 and 1130 and between 1130 and 1335" mean between 720 to 1335? "300-500 ms" – please use the arrow to indicate it, I don't see it.

**Response:** Black arrows have been added in Figure 8 to facilitate interpretation.

28. Row 368: "Figure 9c..." It is quite weird to read directly for 9c, is it the same as 9b? I recommend moving section 5.2 here first.

**Response:** The figure has been updated in the revised manuscript. The interpretation has been added in a new figure (Fig. 10), and all figure references have been corrected accordingly.

29. Row: 370: "unknown faults F1 and F2". I have the impression that these are not unknown faults when looking at Fig. 1. If I roughly extend the marked faults F1 and F2 from Fig. 9c up to the surface, they more or less coincide with the fault in the southern end of the profile LK-11-02 and around the cross-cutting profile LK-11-01 (Fig. 1).

Response: We much appreciate the attention to detail by the reviewer and understand the argumentation. The NNW-trending structures the reviewer refers to in Fig. 1 are low-displacement (< 100m) NNW-trending faults, very common across Ireland (e.g. Moore & Walsh, 2021), and they are largely strike-slip in nature as assessed from densely spaced drill hole constraints at Stonepark and Pallas Green (Blaney and Coffey, 2023). To date, most of the faults in the area have been identified primarily from drilling data, and no major structures of the size of F1/F2 have been positively identified or intersected in the Limerick Syncline (Blaney and Redmond, 2015; Blaney and Coffey, 2023). The F1 and F2 identified in this study represent large displacement Mississippian normal faults relating to basin development, which is inconsistent with the observed map patterns of NNW faults. The existence of F1 and F2 also contrasts strongly with the published models for basin development, e.g. Strogen (1988), Somerville & Strogen (1992), Somerville et al. (1992), who identify the area as representing a carbonate ramp transitioning into a sag basin without any real evidence for a Mississippian syn-rift faulting phase. Therefore, faults F1 and F2 were unknown before. Corroborating this, but not essential

to the argument and not the focus of this study, is that the major Mississippian faults are identified as having a broadly E-W strike, consistent with rift geometries.

30. Row 383: What does "geometries" mean here?

**Response:** This has been removed, and the text modified (rows 362-363).

31. Row 421: I do not see any R1 in Fig. 9? Is it the same as shown in Fig. 8?

**Response:** R1, R2 and R3 are used as references to facilitate the interpretation. The same arrows have also been added to Figure 9 in the revised manuscript.

32. Row 434-439: What is the dominant frequency at the target depth?

**Response:** The dominant frequency is 40 Hz. The information has been added to the text (rows 399-402).

33. Row 453-455: "faults (F1, F2) are now clearly identified" – but earlier in Row 370, it is mentioned they were unknown.

**Response:** The text has been modified accordingly in section 5.4.

34. Row 463: What do you mean by "gentle rotation of the reflectors"?

**Response:** This indicates that the block is slightly rotated due to the segmented fault, with the reflectors exhibiting a gentle dip towards the south. An arrow has been added to the figure to facilitate identification of the feature (Arrow 1; Fig. 10b).

35. Row 464: "domed reflections" – indicate it by arrows.

**Response:** An arrow has been added to the figure to facilitate identification of the feature (Arrow 2; Fig. 10b).

36. Row 477: This section feels redundant given the fact that a very long section 4.1 with the same analogy already exists. I recommend separating them in terms of results and interpretation.

**Response:** This section has been removed from the revised manuscript.

37. Row 485: "Coincidentally with Stonepark" – it is not marked in Fig. 1.

Response: The locations of Stonepark and Pallas Green deposits have been added to Fig. 1.

38. Row 502-503: "We interpreted a south-dipping fault zone, which can be part of the conduits that provided the mineral system with metal-bearing hydrothermal fluids." I think this part has not been properly discussed in the article. The interpretation only comes in the last line in Rows 492-494, where it is from a north-dipping fault, but here it is from a south-dipping fault? Is it because of the syncline structure? Please make sure that this is properly introduced before in the introduction and the geology section.

**Response:** We thank the reviewer for this comment. This interpretation has been added to section 5.3 (rows 413-419).

39. Rows 506-508: Very difficult to follow. Kindly expand it!

**Response:** This has been modified in Section 6, and the conclusions have been significantly sharpened, resulting from the valuable suggestions and discussion points raised by the reviewer.

**Figures**

1. Figure 1: Remove (a) and (b) and just mention that the inset shows a map of Ireland. Remove the other seismic lines other than LK-11-02 as they are not part of this study. The location of Stonepark and Pallas Green is not marked. Increase the font size of drill holes for better visibility. Consider showing a zoom of the northern part of the 2D seismic profile LK-11-02.

**Response:** The figure has been updated. We chose to retain all six 2D profiles, as the datasets are mentioned in the manuscript. To highlight profile LK-11-02, it has been outlined in red on the map.

2. Figure 2: Define what those features are in light pink crosscutting different formations?, Red colour – mineralisation?

**Response:** The figure and caption have been updated.

3. Figure 4: Text size on the axes and legends is too small. Consider replacing the light grey colour with something bright as it is very hard to see.

**Response:** The figure has been updated and is now Figure 5. Abbreviations are used to refer to the units and are kept consistent with the text. The colours remain the same to maintain consistency with the geological map.

4. Figure 5: I am not sure if it is required to show the plot for individual lithologies. At this stage, it was very difficult for me to follow the interpretation of results w.r.t stratigraphic units and individual lithologies, so leave it up to the author(s) to decide, particularly when these individual lithologies are not mapped by the seismic method.

**Response:** This figure has been removed from the revised manuscript. We have added a table summarising the velocities for the main units and have simplified the carbonate textures.

5. Figure 6: I advise enlarging this figure. Remove the single rays going deeper and focus on the upper ~200-300 m where most of the interpretation is done. Use zoom sections on the areas which are discussed in the main text. Add a similar stratigraphic bar on the top as shown in Figure 9c. I also recommend adding a figure with the lithological sections for all the drill holes alongside Table 1. Remember to mark the depths of the boreholes from a common reference datum for a better depth perspective.

**Response:** Figure 6 has been updated accordingly. We have also added a fence diagram of the lithological sections of the drill holes (Fig. 4), as suggested.

6. Figure 7: In (b), I recommend reversing the Y-axis to follow the same depth description as the geological sequence shown in Fig. 2. For (a), similar to Fig. 5, I leave it to the author(s) to decide.

Response: The figure has been updated, and the plot (a) has been removed.

7. Figure 8: It is very difficult to compare the stack section image with the earlier results. The X-axis is defined in terms of CDP points; for Fig. 6, it is in UTM coordinates which is aagin different from Fig. 1. To attain uniformity, I recommend defining the CDP points on the profile in Fig. 1 and use the same CDP point description for Fig. 6. Although the profile length is marked in the bottom axis but then this is not marked in Fig. 1 which again makes it difficult to correlate these results with the geology of the area.

**Response:** We thank the reviewer for this comment. However, the CDP binning does not align precisely with the tomographic section, as the latter is computed in 3D. For this reason, it is not possible to define the CDP points in Fig. 6, as suggested.

8. Figure 9: Move the bar shown in Fig. 9c to the top of the figure. Show amplitude scales for each section. Is 9c the same as 9b with different amplitude scaling? Change the colour for Herberstown

Limestone Fm., it is too light. Dolomitization/Brecciation, Argillaceous, Dykes are either not marked in the figure or are not visible.

**Response:** Figure 9 has been updated to show the comparison between the pre-stack (dataset 2) and post-stack migrations. The interpretation has been added to Figure 10, presenting a depth-converted section along with the seismic interpretation and drill holes. The colours have been retained to maintain consistency with the geological map. Textures and intrusion are indicated in the drill holes according to the legend.

**Tables**

1. Table 2: Please provide the geometry information for both datasets. For dataset 1 at 17 – a dash between 18 and 140.

**Response:** The table has been updated.

2. Table 3: Add the first line mentioning at which step of dataset 2 from Table 2 is used as input data. At stage 5 here, why is the lower end of the frequency chosen as 5 Hz when BP was applied at 12/17 a step before? Please explain.

**Response:** The requested information has been added to the text (row 214). The lower end of the frequency at step 5 was set to 5 Hz to allow a buffer in the FTD noise attenuation, as the bandpass filter applied in the previous step does not necessarily remove all noise.

Bibliography: Please follow the Solid Earth's recommended citation Style. Please leave a gap between citations.

**Response:** The reference list has been revised and formatted in accordance with Solid Earth's citation style.

---

## Referee Report (RR1)

**Dear authors,**

Thanks for the good response and explanations to my comments. All points that were unclear for me have been clarified and the manuscript now sounds smooth and complete to me. The title now is more representative about the study. Objectives and used approach are clearly stated and make easy to follow the work. Following you can find just some minor comment, mostly suggestions.

I really like all used figures, clear and indicative, and you did a good job on improving them, especially figures 4 and 6. Some of the labels are still small (i.e., the axis labels on figure 10), a bit uncomfortable but readable, so it is ok. A suggestion maybe for future works, when you insert the figure in the manuscript your smaller text in the figures should be at least as big as the caption size, in this way you can ensure a clear readability to the reader. I know that some times there is no space for writing bigger text, but you keep them small even when the space was available. Just something to keep in mind for your next work.

I understand the problem of working with legacy data and of its availability, but not having the stacked section from Dataset 2 was a big downside for your work. So, I am glad to see that at the end you managed to use it, it highlights the good results of your work.

In section 3.3 you could add a phrase stating what you said in the response to my comment regarding the starting velocity model used for the tomography "The initial inversions were performed using basic models (1D) with smooth velocity gradients and without introducing lateral velocity variations. These early inversions were primarily aimed at assessing the quality of the picks and identifying potential errors in the acquisition geometry. Even these preliminary inversions yielded excellent convergence, with a significant reduction in misfit (i.e. one order of magnitude). Based on the results of the initial inversions and supported by petrophysical data, smooth lateral velocity variations and variable velocity gradients with depth were introduced in the initial model to reflect trends observed in both datasets. The aim was not to use a highly detailed initial model, but rather to reflect these trends without compromising the independence of the inversion from the initial model. These updated models facilitated faster inversion processes, allowing us to reduce the size of the inversion grid, which resulted in an improvement in the resolution of the final velocity model." You can summarize it but I think it will be interesting and useful to have it in the text.

Line 186. You say that the total record length is 3000 ms, but you are recording a 16 s sweep. So, must likely your total record length is 19 s and 3000 ms after cross correlation with the sweep.

Line 266. The added arrows in figure 6 are very useful, just add in the text "from fig.6" when you refer to them (i.e., arrow 1 in fig. 6).

Figure 10 in the caption you miss a space between "1 and 2". "features identified in the section" is a bit general, I know you describe them properly in the text but the caption should be independent, try to be a bit more specific (i.e., identified features related to faults).

Line 426. I would suggest to replace "have identified a major south-dipping fault (F1) and a north-dipping fault (F2) to the south of Stonepark and Pallas Green." with "have improved the delineation of a major south-dipping fault (F1) and a north-dipping fault (F2) to the south of Stonepark and Pallas Green increasing the confidence on their interpretation" or something similar, since some of these vertical structures (especially the northernmost one) can be inferred already from figure 9a (even if with lower confidence).

I am surprised that in the tomography model there are no low velocity zones corresponding to the identified faulted zones. Could it be because the faults do not reach shallow depth and are deeper than the computed model? Could you discuss it on a sentence?

---

## Author Response (AR2)

Dear Samuel,

Thank you very much for your positive feedback on the revised manuscript and for your additional comments.

Comments and responses below:

In section 3.3 you could add a phrase stating what you said in the response to my comment regarding the starting velocity model used for the tomography "The initial inversions were performed using basic models (1D) with smooth velocity gradients and without introducing lateral velocity variations. These early inversions were primarily aimed at assessing the quality of the picks and identifying potential errors in the acquisition geometry. Even these preliminary inversions yielded excellent convergence, with a significant reduction in misfit (i.e. one order of magnitude).

Based on the results of the initial inversions and supported by petrophysical data, smooth lateral velocity variations and variable velocity gradients with depth were introduced in the initial model to reflect trends observed in both datasets. The aim was not to use a highly detailed initial model, but rather to reflect these trends without compromising the independence of the inversion from the initial model. These updated models facilitated faster inversion processes, allowing us to reduce the size of the inversion grid, which resulted in an improvement in the resolution of the final velocity model.". You can summarize it but I think it will be interesting and useful to have it in the text.

**Response:** Thank you for your suggestion. This has been added to the text (lines 173-183).

Line 186. You say that the total record length is 3000 ms, but you are recording a 16 s sweep. So, must likely your total record length is 19 s and 3000 ms after cross correlation with the sweep.

**Response:** We appreciate the reviewer's observation. We have checked both the raw field data headers and the processing report, and in both sources the record length is specified as 3000 ms. As it is not explicitly stated whether this value refers to post-correlation data, we have retained the information as reported. We have removed "total" from the text (line 188).

Line 266. The added arrows in figure 6 are very useful, just add in the text "from fig.6" when you refer to them (i.e., arrow 1 in fig. 6).

**Response:** This has been modified in the text (lines 268-288).

Figure 10 in the caption you miss a space between "1 and 2". "features identified in the section" is a bit general, I know you describe them properly in the text but the caption should be independent, try to be a bit more specific (i.e., identified features related to faults).

**Response:** This has been modified in the caption.

Line 426. I would suggest to replace "have identified a major south-dipping fault (F1) and a north-dipping fault (F2) to the south of Stonepark and Pallas Green." with "have improved the delineation of a major south-dipping fault (F1) and a north-dipping fault (F2) to the south of Stonepark and Pallas Green increasing the confidence on their interpretation" or something similar, since some of these vertical structures (especially the northernmost one) can be inferred already from figure 9a (even if with lower confidence).

**Response:** We appreciate the reviewer's suggestion. We understand that some of these vertical structures could be inferred from Figure 9a. However, given the significant conceptual geological uncertainty in the study area, the previous interpretations converged on the simplest interpretation

(Bond et al., 2007), which was that of a general dip in the strata. The consensus opinion in published literature and industry reports prior to our work was that the seismic image in Fig. 9a represented the limb of a major syncline (e.g. Fig. 6a in Blaney et al., 2023; Blaney and Redmond, 2015). We have changed 'identified' to 'clearly resolved and delineated with confidence' (line 428), to provide a balanced emphasis on the improved identification of faults with more confidence based on the integrated analysis, while acknowledging the shift in thinking associated with our study.

Bond, C.E., Gibbs, A.D., Shipton, Z.K. and Jones, S., 2007. What do you think this is? "Conceptual uncertainty" in geoscience interpretation. GSA today, 17(11), p.4.

Blaney, D. & Coffey, E. (2023). The volcano-stratigraphic setting of the Pallas Green Zn-Pb deposit, County Limerick. In: Andrew, C.J., Hitzman, M.W. & Stanley, G. 'Irish-type Deposits around the world', Irish Association for Economic Geology, Dublin. 285-308. DOI: https://doi.org/10.61153/QHKU2937

I am surprised that in the tomography model there are no low velocity zones corresponding to the identified faulted zones. Could it be because the faults do not reach shallow depth and are deeper than the computed model? Could you discuss it on a sentence?

**Response:** We appreciate the reviewer's comment. The tomography model shows velocity variations, including low-velocity zones, near the interpreted fault zone F1, close to drill hole TC-2638-088 in Fig. 6. However, the presence of volcanic rocks with variable velocities complicates the direct identification of fault-related velocity anomalies. The tomography model is now mentioned in the revised text (lines 443-444), but a detailed discussion was not included to avoid overinterpretation of these features.